# miR-27b targets MAIP1 to mediate lipid accumulation in cultured human and mouse hepatic cells

Eiko Sakai[1,6], Tsutomu Imaizumi[1,6], Ruruka Suzuki[1], Marcos Taracena-Gándara[1], Toshiki Fujimoto[1], Fuminori Sakurai[1] & Hiroyuki Mizuguchi [1,2,3,4,5✉]

Non-alcoholic liver disease (NAFLD) is a condition caused by excessive fat accumulation in the liver and developed via multiple pathways. miR-27b has been suggested to play crucial roles in the development of NAFLD, assuming via targeting genes involved in lipid catabolism and anabolism. However, other pathways regulated by miR-27b are largely unknown. Here we show that lipid accumulation was induced in miR-27b–transfected human and mouse hepatic cells and that knockdowns of three miR-27b–target genes, β-1,4-galactosyltransferase 3 (B4GALT3), matrix AAA peptidase interacting protein 1 (MAIP1) and PH domain and leucine rich repeat protein phosphatase 2 (PHLPP2), induced lipid accumulation. We also show that B4GALT3 and MAIP1 were direct targets of miR-27b and overexpression of MAIP1 ameliorated miR-27b—induced lipid accumulation. In addition, we show that hepatic Maip1 expression declined in mice fed a high-fat diet, suggesting the involvement of decreased Maip1 expression in the condition of fatty liver. Overall, we identified MAIP1/miR-27b axis as a mediator of hepatic lipid accumulation, a potential therapeutic target for NAFLD.

[1] Laboratory of Biochemistry and Molecular Biology, Graduate School of Pharmaceutical Sciences, Osaka University, 1-6 Yamadaoka, Suita, Osaka 565-0871, Japan. [2] Laboratory of Functional Organoid for Drug Discovery, National Institute of Biomedical Innovation, Health and Nutrition, 7-6-8 Saito, Asagi, Ibaraki, Osaka 567-0085, Japan. [3] Global Center for Advanced Medical Engineering and Informatics, Osaka University, 2-2 Yamadaoka, Suita, Osaka 565-0871, Japan. [4] Integrated Frontier Research for Medical Science Division, Institute for Open and Transdisciplinary Research Initiatives (OTRI), Osaka University, Osaka 565-0871, Japan. [5] Center for Infectious Disease Education and Research (CiDER), Osaka University, Osaka 565-0871, Japan. [6] These authors contributed equally: Eiko Sakai, Tsutomu Imaizumi. ✉email: mizuguch@phs.osaka-u.ac.jp

Non-alcoholic fatty liver disease (NAFLD) is a highly prevalent hepatic manifestation of disordered metabolism, a cluster of conditions that includes obesity, insulin resistance, dyslipidemia and type 2 diabetes[1–4], and 10–30% of NAFLD progresses to non-alcoholic steatohepatitis (NASH)[1,5]. The onset of NAFLD is attributed mainly to hepatic lipid accumulation, and NAFLD leads to NASH, cirrhosis and, eventually, hepatocellular carcinoma. Although the pathogenesis of NAFLD is not fully understood, the current "multiple hit" theory proposes that multiple insults synergistically and in parallel act to induce NAFLD[6,7]. The multiple hits include factors such as insulin resistance[8,9], increased inflammatory cytokines[10,11] and oxidative stress[12–14], circulating adipokines[15,16], intestinal microbiota[17,18], and so on. In addition, crosstalk between them further accelerates progression of disease. However, a full understanding of NAFLD pathogenesis remains to be achieved.

MicroRNAs (miRNAs) are small (19–25 nucleotides long), single-stranded noncoding RNAs that regulate a variety of genes via binding to a perfectly or partially matched complementary sequence located within the 3' untranslated region (UTR) of a target mRNA, resulting in translational suppression and/or mRNA degradation[19–21]. It is now widely recognized that miRNAs regulate a wide variety of biological pathways, including various cellular metabolisms. Liver tissue is rich in miRNAs that have important roles in liver functions[22]. Accmulative studies have demonstrated that several miRNAs regulate lipid metabolism-related genes that are presumably involved in the pathogenesis of NAFLD[23–25]. It was reported that some miRNAs were expressed in serum and liver tissues in good correlation with NAFLD progression[26–29]. The miR-27 family consists of miR-27a and miR-27b, which differ by one base difference at their 3'-terminal regions[30,31]. The upregulated expression of miR-27b was detected in NAFLD patients[28,32]. The miR-27b level in serum was identified as an accurate diagnostic marker for NAFLD. Also, overexpression of hepatitis C virus (HCV) proteins activates miR-27a/b expression; in turn, the overexpression of miR-27a/b in hepatic cell lines results in increased lipid droplets (LDs), suggesting that miR-27a/b expression contributes to the development of hepatic steatosis induced by HCV infection[33]. In animal models, it was demonstrated that miR-27b expression was upregulated in the livers of mice fed a high-fat diet[34]. Together, these previous studies strongly suggest that miR-27b–mediated regulation of lipid metabolism might be a causative factor for NAFLD development.

Hepatic lipid metabolism is tightly regulated by the crosstalk of multiple interrelated pathways. Excess lipid in hepatocytes induces the endoplasmic reticulum (ER) stress which activates altered lipid homeostasis [35–40]. Oxidative stress is induced by impairment of lipid metabolism and in turn, it causes ER stresses and mitochondrial dysfunction which leads to disordered lipid metabolism[41,42]. In silico approach identified miR-27b as a regulatory hub in lipid metabolism due to the presence of predicted target sites in 3'-UTRs and open reading frames (ORFs) of many genes involved in lipid catabolism and anabolism[43]; some of those were experimentally implicated as the direct target of miR-27b[43–47]. Considering a complex manifestation of NAFLD and interrelated regulation of lipid metabolism by multiple pathways, however, it would also be possible that miR-27b contributes to NAFLD development via suppressing genes involved in such pathways, eventually leading to lipid accumulation. In this report, we aimed to uncover another pathway for miR-27b to induce lipid accumulation, the main characteristic of NAFLD, by identifying miR-27b target genes so far unidentified in our knowledge that mediate lipid accumulation.

## Results

**MiR-27b induces lipid accumulation in hepatic cells.** In a previous study, miR-27b and its ortholog miR-27a facilitated lipid accumulation in Huh-7 cells in the context of steatosis induced by HCV infection, where such infection upregulated miR-27a/b expression[33]. To confirm the effects of miR-27b on hepatic lipid accumulation, we transfected Huh-7 cells with miR-27b mimics and evaluated lipid accumulation by labeling lipids with fluorescent dyes, BODIPY 493/503. Lipid accumulation in miR-27b−transfected cells was approximately sevenfold that in control mimic-transfected cells (Fig. 1a and b). Consistent with this result, higher levels of triglyceride (TG) were detected in miR-27b-transfected cells compared to the control when lipids extracted from cells were analyzed (Fig. 1c). Similar phenomena were observed in HepG2 cells transfected with miR-27b mimics (Supplementary Fig. 1a and b). miR-27b mimics also stimulated lipid accumulation in mouse primary hepatocytes (Fig. 1d and e). An increased TG level was detected when miR-27b was transfected, although the difference was not significant ($p > 0.05$) (Fig. 1f). These results suggested that the lipid-related role of miR-27b was not restricted to hepatocellular carcinoma cell lines. Our observations, together with the previous ones[33], revealed the potential of miR-27b to induce lipid accumulation in hepatic cells.

**Functional screening of target genes of miR-27b mediating lipid accumulation.** Next, we aimed to identify genes that mediate lipid accumulation in Huh-7 cells as targets of miR-27b (Fig. 2a). Microarray analyses were performed to detect changes in gene expression in miR-27b-transfected Huh-7 cells. This revealed 749 and 403 genes with twofold up- and down-regulated expression in miR-27b-transfected cells, respectively (Supplementary Data 1 and 2). Functional annotation clustering analyses using DAVID, a functional annotation tool (https://david.ncifcrf.gov/home.jsp), revealed that the up-regulated genes were enriched in multiple pathways with significance (Supplementary Fig. 2a). Enrichment in the hexose metabolic pathway suggests a compensatory activation of the glycolysis pathway for the presumably compromised lipid metabolism (Supplementary Fig. 2a). In fact, several known lipid metabolism-related genes, including fatty acyl CoA reductase 2 (FAR2)[48], glycerol-3-phosphate acyltransferase 2 (GPAT2)[49], apolipoprotein L5 (APOL5)[50], ELOVL fatty acid elongase 4 (ELOVL4)[51] and fatty acid desaturase 2 (FADS2)[52], were up-regulated when individual genes were assessed (Supplementary Table 1). Down-regulated genes were also enriched in multiple pathways, although none of these pathways was related to energy metabolism (Supplementary Fig. 2b). Several lipid-related genes, including acyl-CoA oxidase-like (ACOXL)[53], acyl-CoA dehydrogenase, long chain (ACADL)[54] appeared, however, when the genes were assessed (Supplementary Table 1). ACADL was particularly interesting because it encodes long-chain acyl-CoA dehydrogenase (LCAD), an enzyme that catalyzes the initial step of fatty acid ß-oxidation in mitochondria[54], and the ACADL knockout mice developed more severe fasting-induced hepatic steatosis[55]. Peroxisome proliferator-activated receptor gamma (PPARG) was also interesting because it is a regulator of adipocyte differentiation[56] and is involved in hepatic steatosis[56,57]. Among the down-regulated genes, 85 genes were predicted as human targets of miR-27b according to Targetscan 7.0 (http://www.targetscan.org), a target prediction tool (Supplementary Table 2). Given the conservation of miR-27b across species and the similar effects of miR-27b on mouse hepatocytes to human hepatic cell lines regarding lipid accumulation, it would be highly possible that the human target genes of miR-27b were conserved as target genes in the mouse

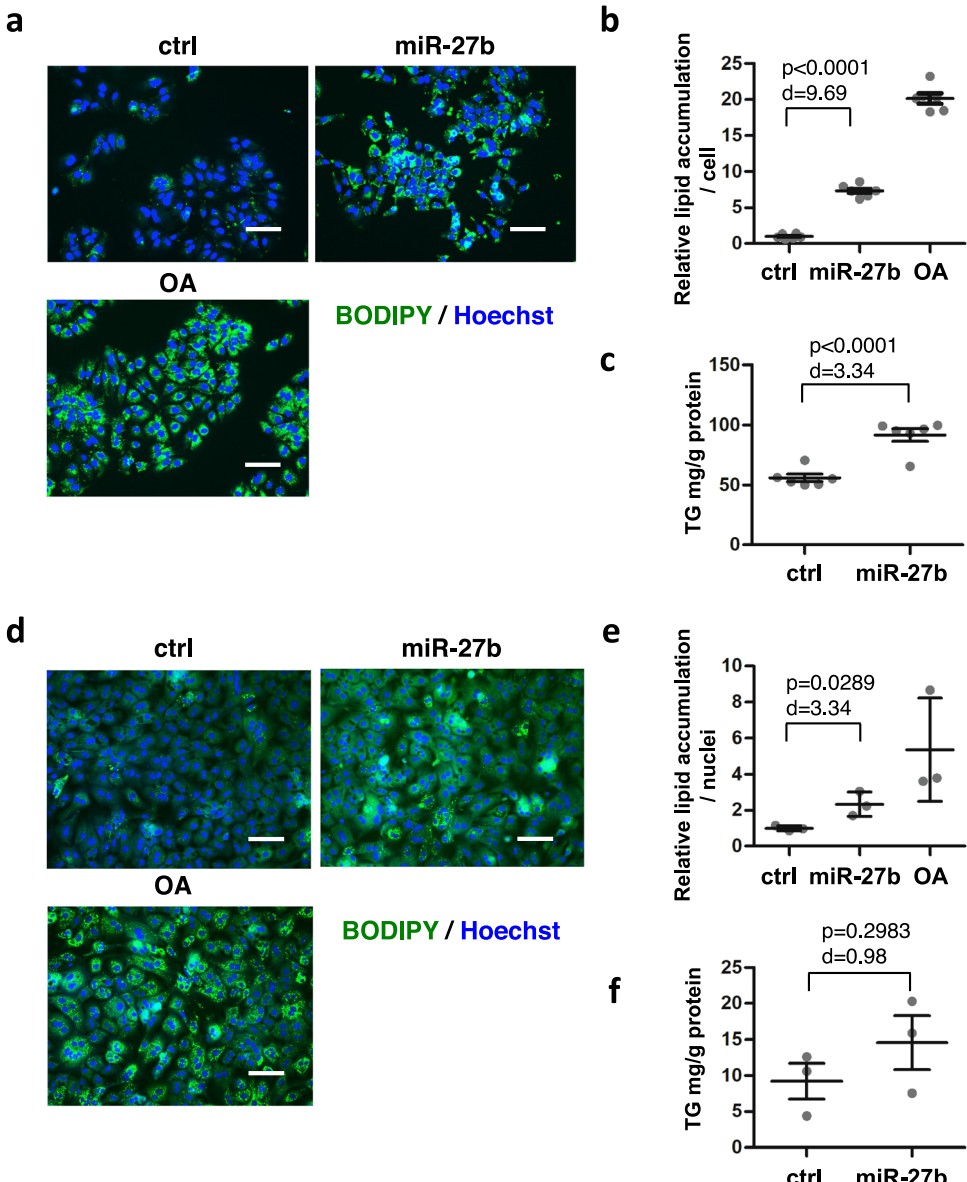

**Fig. 1 *miR-27b* induces lipid accumulation in *mouse* and *human* hepatic cells. a** Huh-7 cells were transfected with control miRNA (ctrl) or miR-27b mimics at 10 nM followed by co-staining with BODIPY493/503 for the lipid and Hoechst 33342 for the nuclei 4 days after transfection. Untransfected cells were treated with oleic acid (OA) at 50 μM for the last 48 h. Scale bars, 100 μm. **b** The relative amount of lipid was estimated based on the images in (**a**). $n = 6$ independent experiments. Results are presented as means ± SE. *p*-values from Student's *t*-test and Cohen's *d* values are shown. **c** Triglycerides (TG) extracted from cells 4 days after transfection were quantitated. $n = 6$ independent experiments. Results are presented as means ± SE. *p*-values from Student's *t*-test and Cohen's *d* values are shown. **d** *Mouse* primary hepatocytes were transfected with the indicated mimics at 50 nM and stained 7 days after transfection as in (**a**). Untransfected cells were treated with OA at 50 μM for the last 48 h. Scale bars, 100 μm. **e** The relative lipid amount was estimated based on the images in (**d**). $n = 3$ biologically independent samples. Representative results from three independent experiments are presented as means ± SD. *p*-value from Student's *t*-test and Cohen's *d* value are shown. **f** TG extracted from cells 7 days after transfection were quantitated. $n = 3$ independent experiments. Results are presented as means ± SE. *p*-value from Student's *t*-test and Cohen's *d* value are shown.

genome. Fifty-two genes were also predicted as miR-27b targets in the *mouse* genome (Supplementary Table 2). Unfortunately, the miR-27b target site in *ACADL* was not conserved in the *mouse* genome. *PPARG*, a known direct target of *miR-27b*[43,45,46] was included in the list. However, hepatocyte-specific knockout experiments as well as overexpression experiments revealed that the hepatic *PPARG* expression positively regulated hepatic lipid accumulation[58–60], suggesting that suppression of *PPARG* by *miR-27b* would reduce lipid accumulation. Therefore we decided to continue the screening, with *PPARG* excluded from it, for genes that mediate lipid accumulation when suppressed. We

examined the remaining 51 genes for target site conservation and expression by qRT-PCR. The *miR-27b*–target sites in 32 out of the 51 genes were conserved in the *mouse* genome (Supplementary Table 3). Four out of the 32 genes could not be analyzed for expression presumably due to their low expression, while twelve genes were suppressed by *miR-27b* to below 60% of control mimics (Fig. 2b and Supplementary Table 3). Noticeably, *cyclin G1 (CCNG1)*, a previously reported target gene of *miR-27b* as a negative regulator of p53 in various types of cancer cells[61], was included, indicating that the screening was reliable. The remaining 11 genes were further examined.

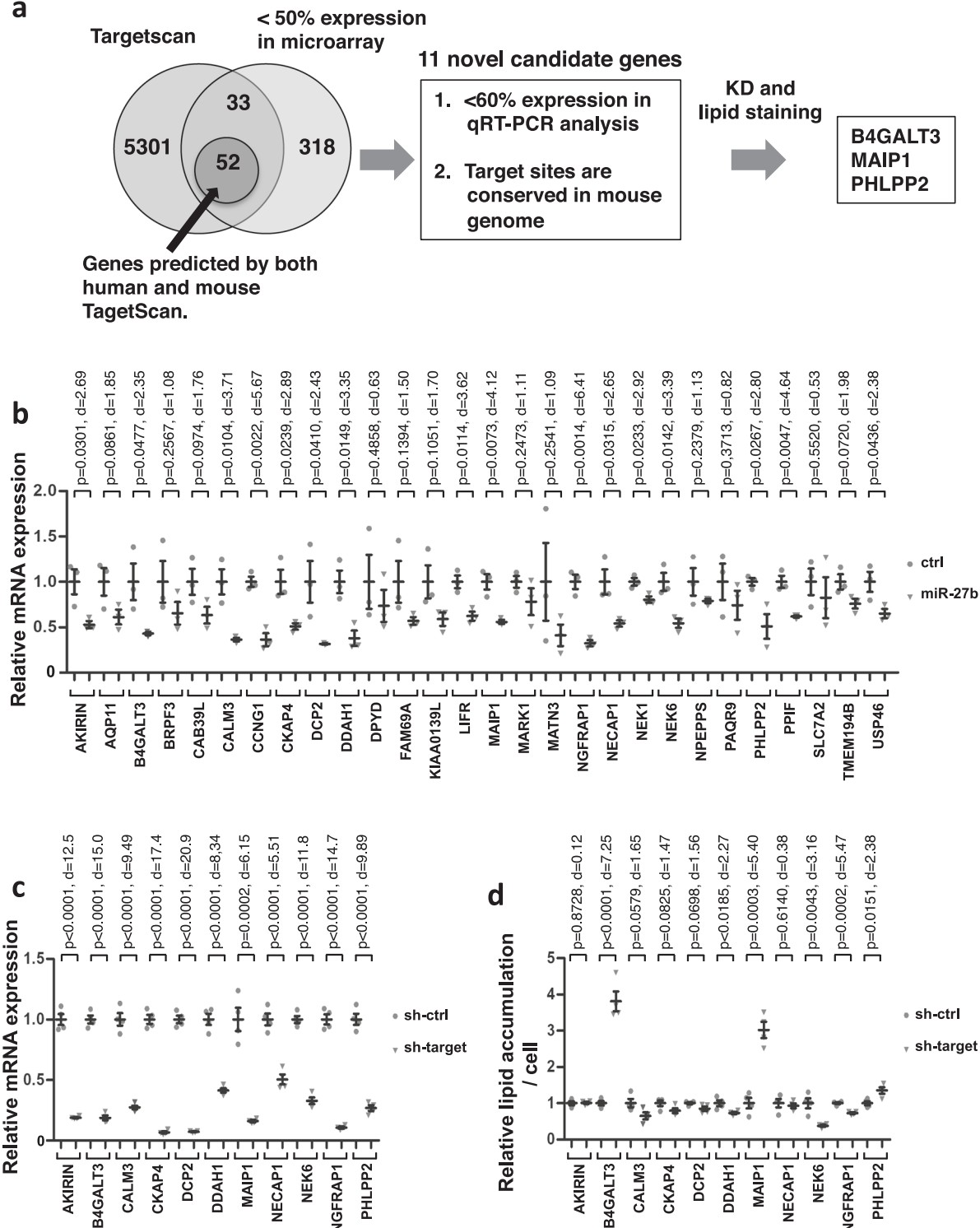

To explore genes related to lipid metabolism, the expression of the 11 genes was individually suppressed by shRNAs for each of the candidate genes, and the lipid accumulation was evaluated in Huh-7 cells. The knockdown efficiency was confirmed (Fig. 2c). Then, lipid accumulation was assessed by staining cells with BODIPY493/503, demonstrating that the knockdown of three genes, *β-1,4-galactosyltransferase 3* (*B4GALT3*), *matrix AAA peptidase interacting protein 1* (*MAIP1*, previously known as *C2orf47*) and *PH domain and leucine rich repeat protein phosphatase 2* (*PHLPP2*), enhanced lipid accumulation (Fig. 2d).

**Knockdown of *B4GALT3* and *MAIP1* induces lipid accumulation.** To confirm that the decreased expression of these three genes was responsible for lipid accumulation, we performed knockdown experiments using siRNAs (Fig. 3). Huh-7 cells were transfected with siRNAs for the three genes and the knockdown efficiencies were assessed, confirming that they were efficiently suppressed (Fig. 3a and b). Under these conditions, lipid accumulation was evaluated, demonstrating that lipid accumulation was stimulated by knockdown of *B4GALT3* and of *MAIP1*. The knockdown of *PHLPP2* also caused an increase in lipid

**Fig. 2 Screening of miR-27b target genes by in silico analyses and knockdown experiment. a** The schema for identifying the *miR-27b* target genes. In silico target prediction and microarray analysis identified 85 genes as target gene candidates. The *mouse* orthologs of 52 of the 85 genes were also predicted as *miR-27b* target genes. Then 32 of the 52 genes were further validated for target site conservation in *human* and *mouse* and for suppression by *miR-27b* mimics using qRT-PCR. Eleven genes were finally identified as candidate genes that *miR-27b* suppressed to below 60 % and that had conserved target sites between *human* and *mouse*. Knockdown experiments were carried out and lipid accumulation was evaluated, raising the three genes as the candidates. **b** Huh-7 cells were transfected with control miRNA (ctrl) or *miR-27b* mimics at 10 nM and mRNA expression of the candidate genes with conserved target sites was analyzed. $n = 3$ independent experiments. Results are presented as means ± SE. *p*-values from Student's *t*-test and Cohen's *d* values are shown. Gray circles, ctrl; gray triangles, *miR-27b*. **c** Huh-7 cells were transduced with lentiviral (LV) vectors expressing an shRNA for each candidate gene at 30 MOI. mRNA was prepared 3 days after transduction and mRNA levels of the target genes were analyzed by qRT-PCR. Relative mRNA expression levels are shown compared to those when the cells were transduced with the control shRNA-expressing LV vector, pLKO.1-puro-shRNA-control. $n = 4$ independent experiments. Results are presented as means ± SE. *p*-values from Student's *t*-test and Cohen's *d* values are shown. Gray circles, sh-ctrl; gray triangles, sh-target. **d** Lipids were stained with Bodipy493/503 4 days after transduction with LV vectors, and lipid amounts were evaluated based on the images. The relative lipid amount compared to the control LV vector was shown. $n = 4$ independent experiments. Results are presented as means ± SE. *p*-values from Student's *t*-test and Cohen's *d* values are shown. Gray circles, sh-ctrl; gray triangles, sh-target. sh-ctrl, LV vector expressing the control shRNA; sh-target, LV vectors expressing shRNAs for target genes.

accumulation (Fig. 3c and d). Similarly, higher levels of triglyceride (TG) were detected in cells transfected with siRNAs for *B4GALT* and *MAIP1* compared to the control. Transfection of cells with si-*PHLPP2* also resulted in higher levels of TG (Fig. 3e).

**MiR-27b directly regulates expression of B4GALT3 and MAIP1, but not PHLPP2.** Next we examined whether *B4GALT3*, *MAIP1*, and *PHLPP2* were actually regulated by *miR-27b*. Huh-7 cells were transfected with *miR-27b* mimics, and mRNA and protein expressions were analyzed. The mRNA expression levels of all three genes were suppressed by ectopic expression of *miR-27b*, which was well consistent with the results obtained in Fig. 2b (Fig. 4a). Similar suppression was observed in HepG2 cells transfected with *miR-27b* mimics (Supplementary Fig. 1c). Suppression of the protein expression was also confirmed (Fig. 4b and c). Further, expression of *Maip1* and *B4galt3* was suppressed in *mouse* primary hepatocytes when transfected with *miR-27b* mimics (Fig. 4d). These results clearly indicated that *miR-27b* suppressed the expression of *MAIP1* and *B4GALT3*, and presumably *PHLPP2*, both in *human* and *mouse* hepatic cells.

Since these three genes have the target sequences, it is highly possible that they were directly suppressed by *miR-27b*. Therefore, we examined the direct interaction between *miR-27b* and the 3'UTRs of the three targets by a reporter assay (Fig. 4e). Two *miR-27b* target sequences were predicted in the 3'UTR of *B4GALT3* and of *MAIP1*, while three were predicted in the 3'UTR of *PHLPP2*. The 3'UTRs were cloned into the reporter plasmids downstream of the *luciferase* genes, and mutations were introduced in the target sites. Huh-7 cells were co-transfected with *miR-27b* mimics and the wild-type or mutant reporter plasmids. *miR-27b* suppressed the *luciferase* gene expressions from wild-type reporter plasmids of *B4GALT3* and *MAIP1*. Conversely, the suppressions were canceled when mutations were introduced in all the target sites. Meanwhile, the suppressive effect of *miR-27b* on the *PHLPP2* reporter plasmid was relatively mild and the effects of the mutations were subtle. These results suggested that *miR-27b* suppressed *B4GALT3* and *MAIP1* directly, and *PHLPP2* rather in an indirect manner.

Taken together, these results clearly demonstrated that *miR-27b* reduced the expression levels of *B4GALT3* and *MAIP1* directly via binding to the target sequences in their 3'UTRs.

**Overexpression of MAIP1 ameliorated miR-27b-induced lipid accumulation.** As demonstrated above, a decrease in the *B4GALT3* and *MAIP1* expression levels caused lipid accumulation and these genes were direct targets of *miR-27b*. Therefore, we examined whether *miR-27b* induces lipid accumulation via the suppression of *B4GALT3* and *MAIP1* expression (Fig. 5). Huh-7 cells were transduced with LV vectors expressing *B4GALT3* or *MAIP1* and then transfected with *miR-27b* mimics. The expression levels of *B4GALT3* and *MAIP1*, which *miR-27b* suppressed, were restored to approximately two to threefold the endogenous expression levels by LV vector-mediated overexpression (Fig. 5a and d). Under these conditions, the overexpression of *MAIP1* partially repressed the *miR-27b*-induced lipid accumulation (Fig. 5b and c). The overexpression of *MAIP1* slightly repressed the TG level compared to the LV-ctrl (Fig. 5d). On the other hand, overexpression of *B4GALT3* did not cancel the lipid accumulation induced by *miR-27b* (Fig. 5e–h).

Taken together, these results indicated that *MAIP1* was a direct target of *miR-27b* and mediated, at least in part, *miR-27b* −induced lipid accumulation.

**Decreased expression of MAIP1 did not inactivate AMP-activated protein kinase (AMPK).** Next, we attempted to elucidate the pathway by which *MAIP1* affected lipid accumulation. Previous studies reported that *essential MCU regulator (EMRE)*, a component of the calcium uniporter complex that includes the *mitochondrial calcium uniporter (MCU)*, was assisted by *MAIP1* to localize to the mitochondrial membrane[62] and that hepatocyte-specific defects of *MCU* induced steatosis in mice, which was implicated in inactivation of *Ampk*, a master regulator of energy expenditure, in a cytoplasmic Ca2+-dependent manner[63]. Therefore, it was possible that *MAIP1* depletion induced inactivation of *AMPK*, which could have been the cause of the lipid accumulation. To test this hypothesis, *MAIP1* was knocked down in Huh-7 cells and the phosphorylated state of *AMPK* was examined. However, neither the phosphorylated protein level nor the phosphorylated and total *AMPK* protein ratio was suppressed by the depletion of *MAIP1* protein, suggesting that *AMPK* was not the mediator of lipid accumulation by *MAIP1* depletion (Supplementary Fig. 3).

**Decreased expression of Maip1 induced lipid accumulation in primary hepatocytes and was observed in the livers of HFD-fed mice.** Next, we evaluated the association between *mouse Maip1* expression and hepatic lipid accumulation. To examine the effect of decreased *Maip1* expression on liver lipid accumulation, primary *mouse* hepatocytes were treated with siRNA for *Maip1*. Moderately down-regulated expression of Maip1 was confirmed (Fig. 6a). Under these conditions, lipid accumulation was stimulated in hepatocytes treated with si-*Maip1*, suggesting that down-regulated expression of Maip1 induces lipid accumulation in primary hepatic cells as observed in Huh-7 cells (Fig. 6b and c). The TG level revealed a upward trend when si-*MAIP1* was

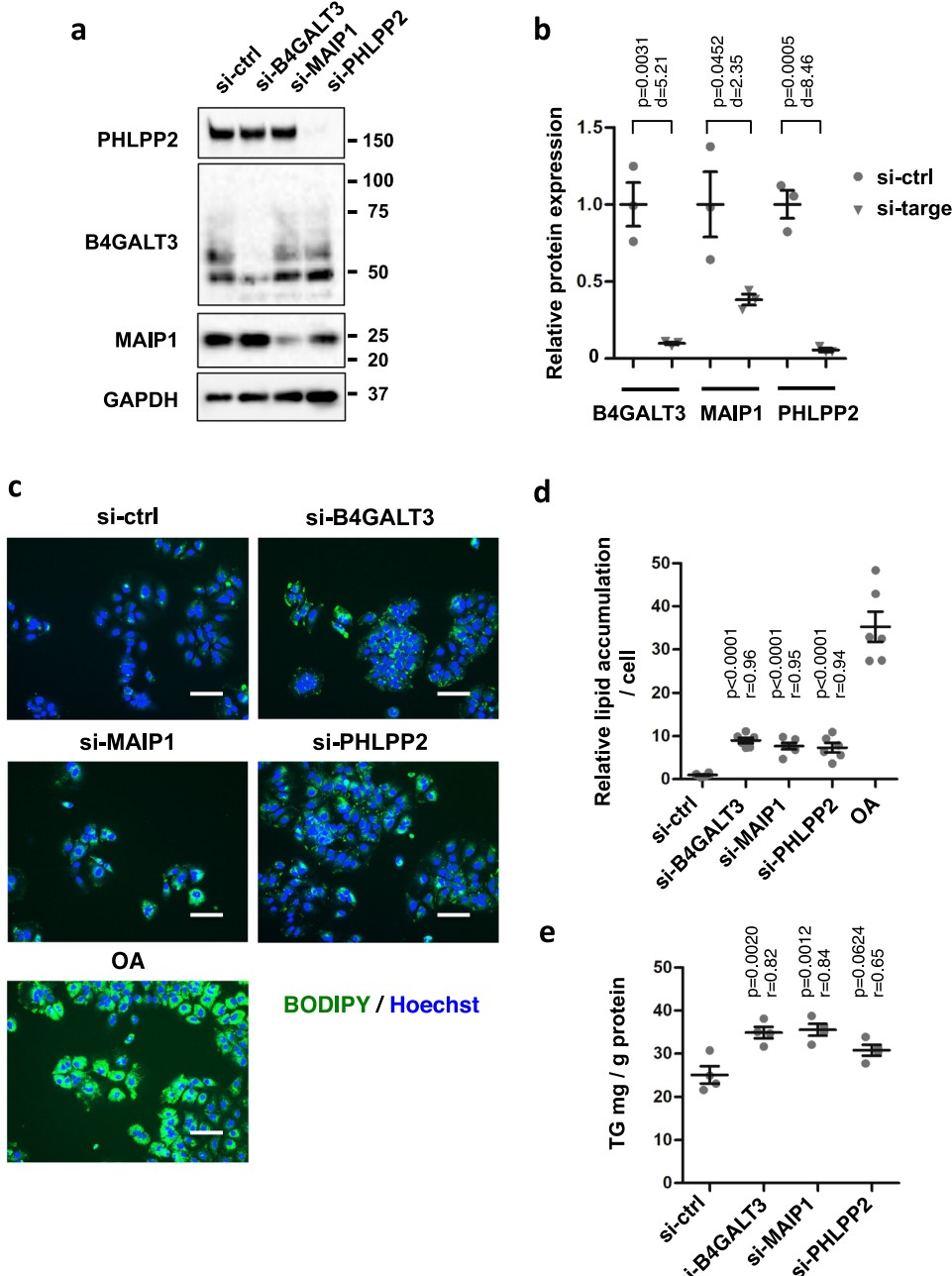

**Fig. 3 Knockdowns of *B4GALT3*, *MAIP1*, and *PHLPP2* induce lipid accumulations in Huh-7 cells. a** Huh-7 cells were transfected with control siRNA (si-ctrl) or siRNA for each gene at 10 nM. Protein expression was examined 3 days after transfection by Western blotting. Molecular weight markers are shown on the right. *GAPDH* is for the internal control. **b** Protein expression levels were evaluated based on the images in (**a**). n = 3 independent experiments. Results are presented as means ± SE. *p*-values from one-way ANOVA followed by Bonferroni's post-hoc comparisons tests for si-ctrl and Pearson's *r* values are shown. si-ctrl, negative control siRNA; si-target, the corresponding siRNA for the target gene analyzed. Gray circles, si-ctrl; gray triangles, si-target. **c** Huh-7 cells were transfected with siRNAs at 10 nM. Cells were co-stained with BODIPY493/503 for the lipid and Hoechst 33342 for the nuclei 4 days after transfection. Untransfected cells were treated with oleic acid (OA) at 50 μM for the last 48 h. Scale bars, 100 μm. **d** The amount of lipids was evaluated based on the images in (**c**). The relative lipid accumulation of each sample was compared to that of the control siRNA. n = 6 independent experiments. Results are presented as means ± SE. **e** Triglycerides (TG) extracted from cells 4 days after transfection were quantitated. n = 4 independent experiments. Results are presented as means ± SE. *p*-values from one-way ANOVA followed by Bonferroni's post-hoc comparisons tests for si-ctrl and Pearson's *r* values are shown in (**d**) and (**e**).

transfected (Fig. 6d). The high-fat diet-fed *mouse* is a widely used animal model for fatty liver diseases. In order to examine the *Maip1* expression in fatty livers, C57B/6 mice were fed either a basic or high-fat diet for 8 and 16 weeks followed by an analysis of the *Maip1* expression. A slight decline in *Maip1* expression was observed in the livers of HFD-fed mice at 8 weeks of feeding

(Fig. 6e), and this decline became more prominent at 16 weeks of HFD feeding (Fig. 6f), suggesting that *Maip1* expression decreased in association with the fatty liver progression. Of note, similar reductions in *B4galt3* and *Phlpp2* expression were observed, indicating that these expressions were also associated with in vivo fatty liver conditions.

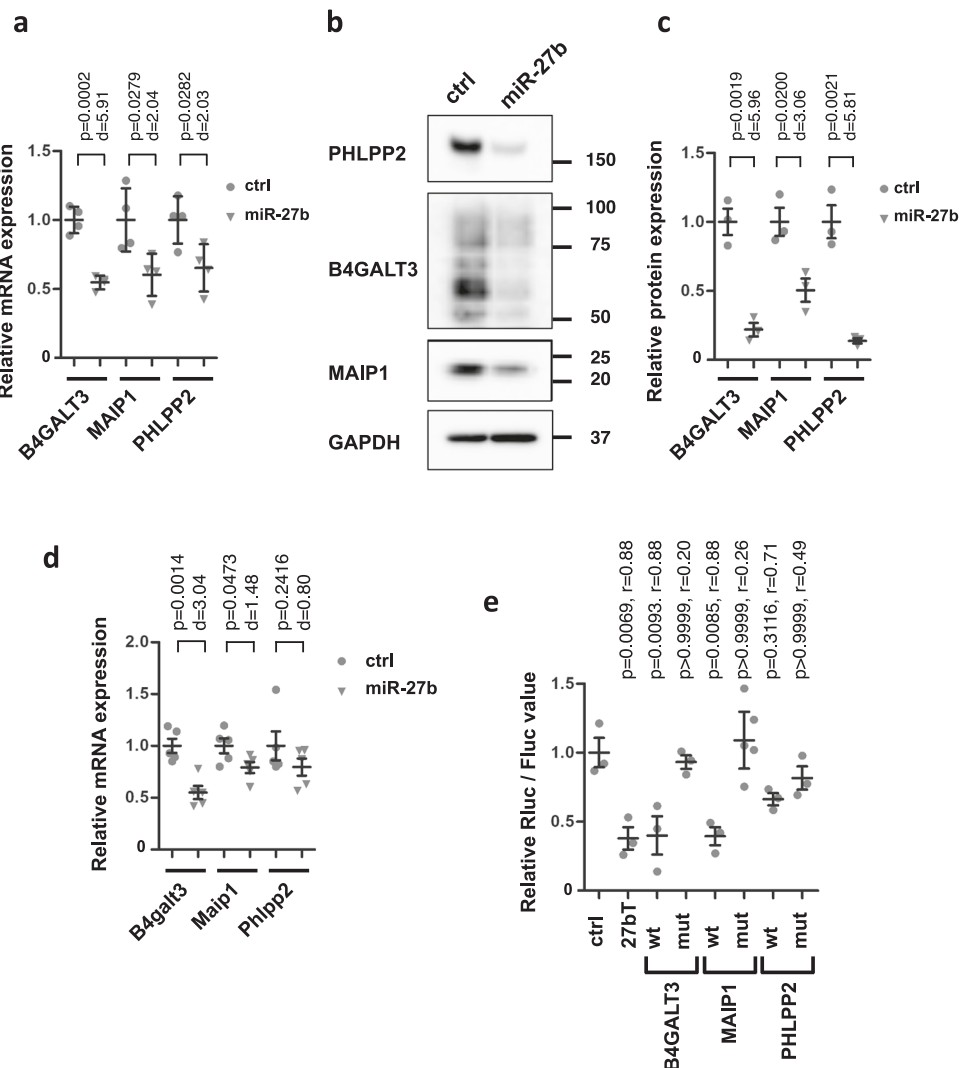

**Fig. 4 miR-27b suppresses the expression of B4GALT3 and MAIP1 as direct targets. a** Huh-7 cells were transfected with control miRNA (ctrl) and miR-27b mimics at 10 nM, and mRNA levels were examined 2 days after transfection by qRT-PCRs. n = 4 biologically independent samples. Representative results from three independent experiments are shown. Results are presented as means ± SD. p-values from Student's t-test and Cohen's d values are shown. Gray circles, ctrl; gray triangles, miR-27b mimics. **b** Protein expression was examined 3 days after transfection by Western blotting. Molecular weight markers are shown on the right. GAPDH is for the internal control. **c** Quantitation of protein expression. n = 3 independent experiments. Results are presented as means ± SE. p-values from Student's t-test and Cohen's d values are shown. Gray circles, ctrl; gray triangles, miR-27b mimics. **d** Primary mouse hepatocytes were transfected with control miRNA (ctrl) and miR-27b mimics at 50 nM, and mRNA levels were examined 2 days after transfection by qRT-PCRs. n = 5 independent experiments. Results are presented as means ± SE. p-values from Student's t-test and Cohen's d values are shown. Gray circles, ctrl; gray triangles, miR-27b mimics. **e** Huh-7 cells were co-transfected with reporter plasmids containing wild-type (wt) or mutated (mut) 3'UTRs with control miRNA and miR-27b mimics (10 nM). The empty reporter plasmid and the reporter plasmid containing the complementary sequence to miR-27b were used as a negative (ctrl) and a positive (27bT) control, respectively. Luciferase assay was performed 2 days after transfection. The renilla luciferase (Rluc) activity was normalized by firefly luciferase (Fluc) activity. The relative Rluc/Fluc values of miR-27b transfectants to the control miRNA mimic transfectants are shown. n = 3 independent experiments. Results are presented as means ± SE. The relative Rluc/Fluc of each reporter plasmid was compared to that of the control reporter plasmid. p-values from one-way ANOVA followed by Bonferroni's post-hoc comparisons tests for ctrl and Pearson's r values are shown.

**Gene expression analysis of MAIP1, B4GALT3 and PHLPP2 in liver biopsies form NASH patients.** We also performed a separate analysis using a public database of RNA-seq data from the livers of NASH patients (GEO, GE225740)[64,65], in which we searched for the expressions of MAIP1, B4GALT3, and PHLPP2 (Supplementary Fig. 4). The RNA samples were categorized into three groups (0, no steatohepatitis; 1, borderline steatohepatitis; 2, definite steatohepatitis) according to Powell et al.[65]. The expression of MAIP1 tended to decline with increasing level of steatohepatitis, suggesting that MAIP1 was involved in the manifestation.

Taking these results together, we identified a miR-27b-mediated regulatory pathway that targets MAIP1 and that eventually leads to hepatic lipid accumulation, as illustrated in Fig. 6g.

## Disccussion

NAFLD is characterized by excessive lipid accumulation in the liver. A growing body of evidence suggests that miR-27b is involved in hepatic steatosis by altering lipid metabolisms. In this study, we screened for miR-27b target genes, whose knockdowns induced lipid accumulation, and identified B4GLAT3, MAIP1,

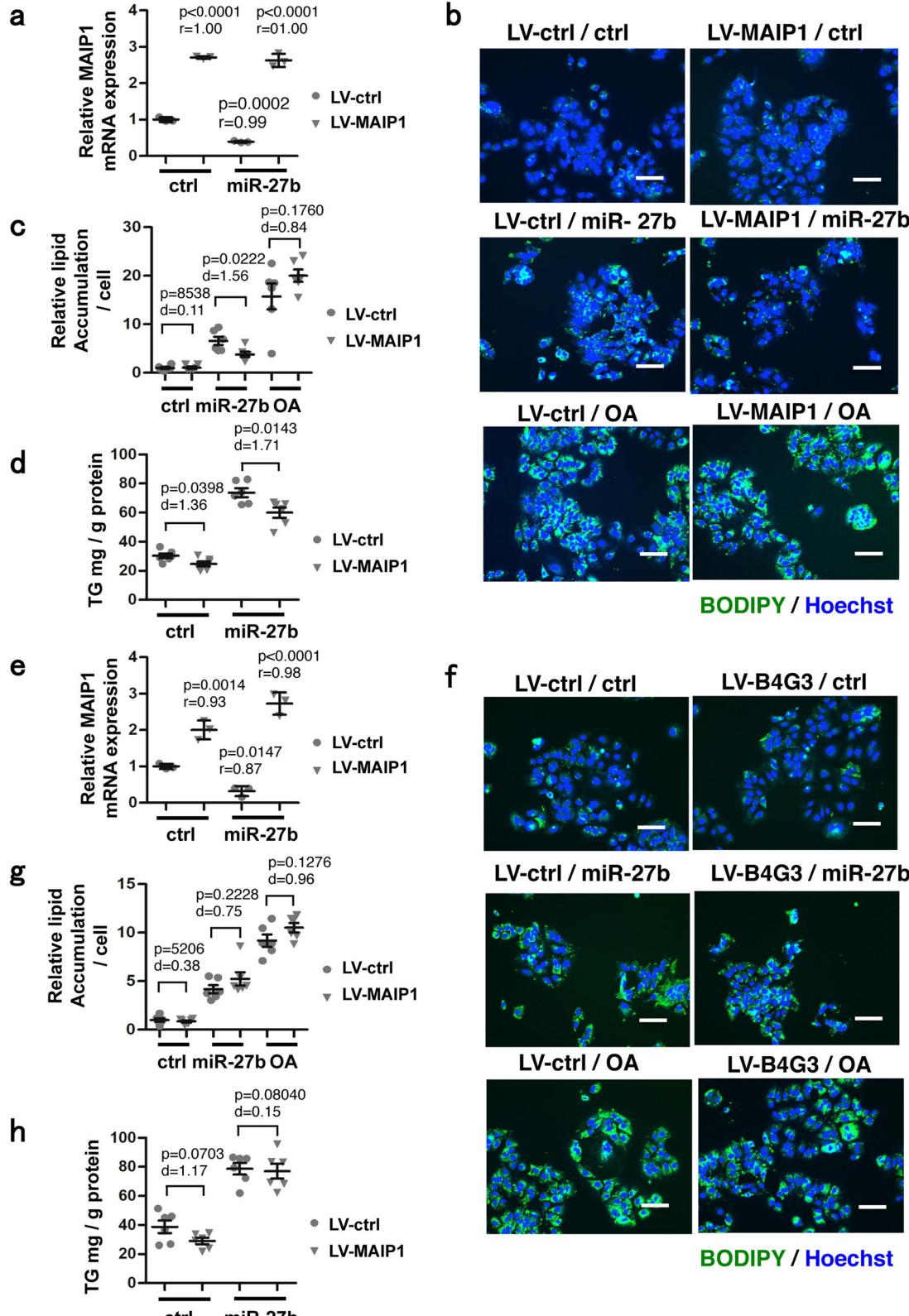

and *PHLPP2*. *PHLPP2* has been implicated in repressing the *AKT serine/threonine kinase (AKT)/mechanistic target of rapamycin complex (mTORC) 1* pathway, which regulates de novo lipogenesis, as discussed below[66–68]. However, *B4GALT3* and *MAIP1* have not been directly implicated as regulators of lipid accumulation. Therefore, the latter two genes seem to be yet identified effectors whose altered expression induces lipid accumulation.

*B4GALT3* is a resident of the Golgi apparatus in cells, which is where glycosylation processes occur and where the glycosylated proteins are then sorted. *B4GALT3*, one of seven *β-1, 4-galactosyltransferases*, catalyzes the synthesis of the first N-acetyllactosamine unit of poly-N-acetyllactosamine chains and further glycosylates N-linked glycosylated proteins transported from ER[69,70]. Integrin *β1* is so far the only reported substrate of

**Fig. 5 Restoring MAIP1-expression ameliorates the lipid accumulation induced by *miR-27b*. a** Huh-7 cells were transduced with LV-vectors expressing *MAIP1* (20 MOI) followed by transfection with control miRNA (ctrl) and *miR-27b* mimics (10 nM) 24 h later. The mRNA levels of *MAIP1* were analyzed 48 h after transfection by qRT-PCRs, and the relative expression of *MAIP1* was evaluated. $n = 3$ biologically independent samples. Representative results from three independent experiments are presented as means ± SD. *p*-values from one-way ANOVA followed by Bonferroni's post-hoc comparisons tests for LV-ctrl-transduced and ctrl-transfected samples and Pearson's *r* values are shown. Gray circles, LV-ctrl; gray triangles, LV-MAIP1. **b** Huh-7 cells transduced with LV-vectors expressing *MAIP1* (20 MOI) were transfected with control miRNA (ctrl) or *miR-27b* mimics (10 nM) and co-stained with BODIPY493/503 for the lipid and Hoechst 33342 for the nuclei 4 days after transfection. Untransfected cells were treated with oleic acid (OA) at 50 μM for the last 48 h. Scale bars, 100 μm. **c** The relative amount of lipid was evaluated based on the images in (**b**). $n = 6$ independent experiments. Results are presented as means ± SE. *p*-values from Student's *t*-test and Cohen's *d* values are shown. Gray circles, LV-ctrl; gray triangles, LV-MAIP1. **d** Triglycerides (TG) extracted from cells 4 days after transfection were quantitated. $n = 6$ independent experiments. Results are presented as means ± SE. *p*-values from Student's *t*-test and Cohen's *d* values are shown. Gray circles, LV-ctrl; gray triangles, LV-MAIP1. **e** Huh-7 cells were transduced with LV-vectors expressing *B4GALT3* (50 MOI) followed by transfection with control miRNA (ctrl) and *miR-27b* mimics (10 nM) 24 h later. The mRNA levels of *B4GALT3* were evaluated. $n = 3$ biologically independent samples. Representative results from three independent experiments are presented as means ± SD. *p*-values from one-way ANOVA followed by Bonferroni's post-hoc comparisons tests for LV-ctrl–transduced and ctrl–transfected samples and Pearson's *r* values are shown. Gray circles, LV-ctrl; gray triangles, LV-B4GALT3. **f** Huh-7 cells transduced with LV-vectors expressing *B4GALT3* (50 MOI) were transfected with control miRNA (ctrl) or *miR-27b* mimics (10 nM) and co-stained with BODIPY493/503 and Hoechst 33342 4 days after transfection as in (b). Untransfected cells were treated with OA at 50 μM for the last 48 h. Scale bars, 100 μm. **g** The relative amount of lipid was evaluated based on the images in (**f**). $n = 6$ independent experiments. Results are presented as means ± SE. *p*-values from Student's *t*-test and Cohen's *d* values are shown. Gray circles, LV-ctrl; gray triangles, LV-B4GALT3. **h** TG extracted from cells 4 days after transfection were quantitated as in (**d**). $n = 6$ independent experiments. Results are presented as means ± SE. *p*-values from Student's *t*-test and Cohen's *d* values are shown. Gray circles, LV-ctrl; gray triangles, LV-B4GALT3. LV-ctrl, control LV vector; LV-MAIP1, *MAIP1*-expressing LV vector; LV-B4G3, *B4GALT3*-expressing LV vector.

B4GALT3, and impaired expression of *B4GALT3* in tumors causes modifications in the behavior of cancer cells[69,71,72]. Proteins critical for lipid metabolism, such as lipases and proteins involved in LD formation, are the substrates for N-glycosylation[69,73,74]. Besides compromising the glycosylation of lipid metabolism-related proteins, disrupted glycosylation interrupts protein folding in the ER, inducing the accumulation of misfolded proteins in the ER, a condition of ER stress[75,76]. ER stress stimulates LD synthesis through the unfolded protein response (UPR) by activating ER stress sensors, *endoplasmic reticulum to nucleus signaling 1* (IRE1a), *eukaryotic translation initiation factor 2α kinase 3* (PERK), and *activating transcription factor 6* (ATF6)[35-40]. In a yeast study, James et al. demonstrated that N-glycosylation deficiency induced ER stress accompanied by increased levels of LDs[77]. Further studies are required to elucidate the regulatory mechanisms of lipid accumulation by *B4GALT3*.

MAIP1 has been reported to be a mitochondrial matrix protein and implicated in protecting EMRE, a component of the calcium uniporter complex, *MCU*, from proteolysis by assisting membrane insertion[62]. Mitochondrial calcium uptake is involved in the regulation of a variety of cellular bioenergetic events[78-80]. Tomar et al. reported that a liver-specific *MCU* knockout in mice caused an increase in hepatic lipid accumulation, which was implicated as a result of an elevated cytoplasmic calcium level[63]. An increased cytoplasmic calcium level activates Ca-dependent protein phosphatase 4 (PP4) which, in turn, dephosphorylates and thus inactivates AMP-activated protein kinase (AMPK), resulting in decreased energy expenditures[63]. In addition, the impairment of *MCU* functions, which might be caused by *MAIP1* depletion, may directly delay energy consumption, that is to say, suspended fatty acid oxidation, since oxidative phosphorylation depends on mitochondrial Ca2+[81,82]. Although neither decreased Ca2+ influx into mitochondria nor any changes in the cytoplasmic Ca2+ level were detected in the *MAIP1*-depleted non-hepatic cell line in the previous report[62], such changes involving Ca2+ might still occur in hepatic cells. However, inactivation of AMPK was not observed in *MAIP1*-depleted Huh-7 cells (Supplementary Fig. 2), suggesting that massive changes in Ca2+ influx into mitochondria, such as would be sufficient to change the cytoplasmic Ca2+ levels, also did not occur by knockdown of *MAIP1* in hepatic cells.

Alternatively, the localization of enzymes involved in β-oxidation to the mitochondrial membrane might require the assistance of MAIP1, similarly to the case with EMRE; thus, depletion of *MAIP1* might disturb fatty acid degradation, in turn inducing lipid accumulation[62]. It is also possible that *MAIP1* would be required to maintain normal mitochondrial functions, probably via assisting a wide variety of mitochondrial proteins to be transported into mitochondria or integrated into the mitochondrial membrane; thus *MAIP1*-depletion would induce overall mitochondrial dysfunctions with increased oxidative stresses which might cause the lipid accumulation[42]. From the pathological point of view, morphological abnormality in mitochondria was observed in the livers of NASH patients[83] and mitochondrial dysfunction was suggested to play a central role in the development and progression of NAFLD[84-87]. These possibilities, in particular the contribution of *MAIP1*-depletion to the development of NAFLD, need to be elucidated in future studies.

*PHLPP2* is a regulator of *AKT* and *protein kinase C* (PKC) together with *PHLPP1* and has been reported to play suppressive roles in cancers[88]. *PHLPP2* also has been implicated in terminating insulin signaling by dephosphorylating *AKT* and thus inactivating *mTORC1*[67]. *PHLPP2* knockout in hepatoma cells actually demonstrated prolonged effects on insulin signals[68]. Since insulin generally stimulates de novo lipogenesis through the *AKT/mTORC1* pathway in the liver[89,90], a decrease in *PHLPP2* expression might induce accelerated lipogenesis, leading to increased lipid accumulation. In fact, Kim et al. have reported that liver-specific *PHLPP2*-knockout mice exhibited hepatic steatosis[68]. They also provided evidence that the degradation of *PHLPP2* by the ubiquitin pathway was involved in the down-regulated expression of *PHLPP2* in NAFLD[68]. Although the extent of lipid accumulation caused by *PHLPP2* knockdown was not large in the short periods in our knockdown experiments (Fig. 3), our observation that *miR-27b* overexpression clearly suppressed *PHLPP2* expression (Fig. 4) presents *miR-27b* as an unidentified player, in our knowledge, in the *AKT/mTORC1*-dependent network of lipid metabolisms.

In conclusion, our study provides additional insights into the molecular mechanism underlying hepatic steatosis caused by *miR-27b* and identifies *B4GALT3* and *MAIP1* as potential therapeutic targets for NAFLD.

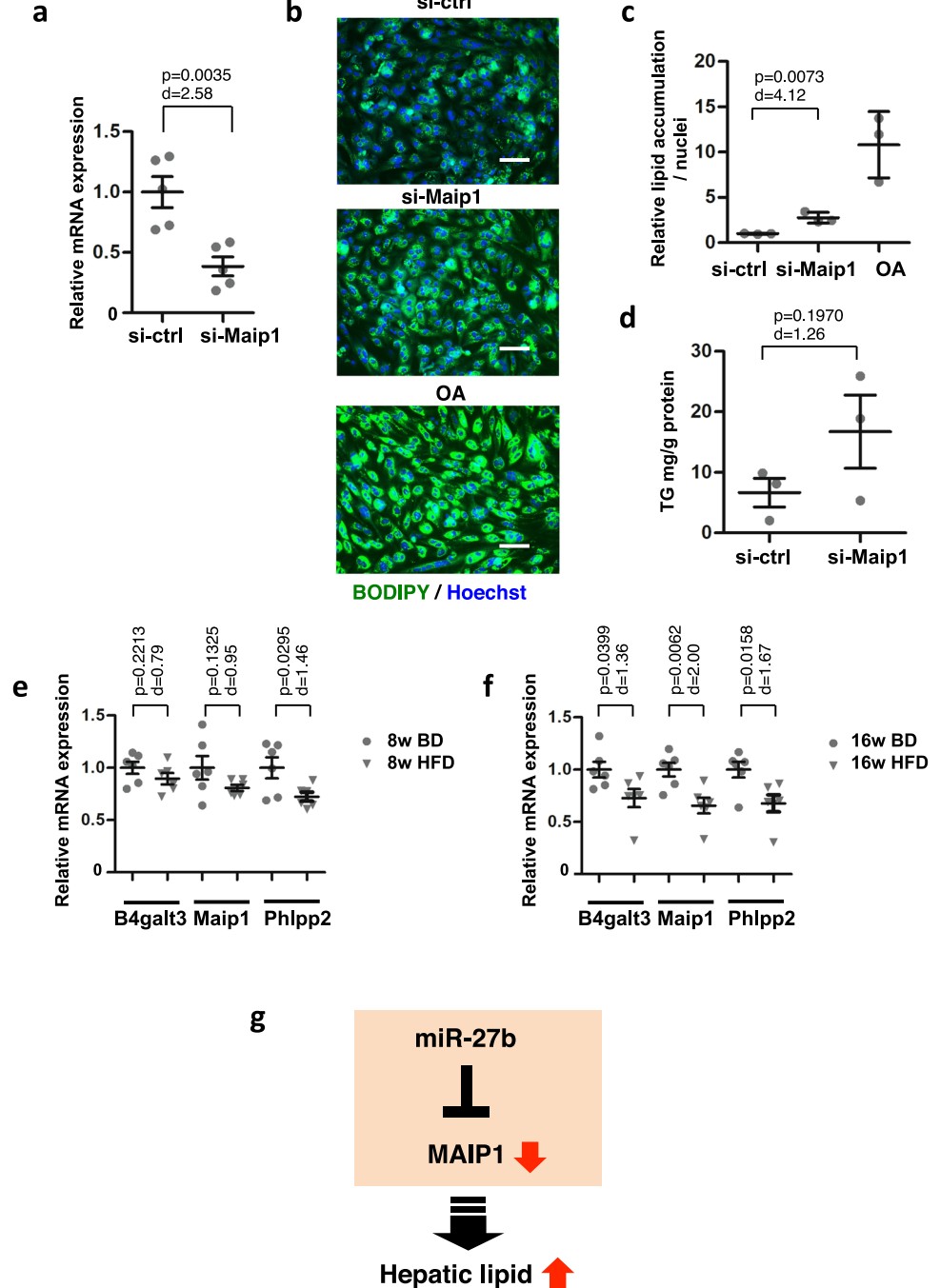

**Fig. 6 Decreased expression of *Maip1* is associated with lipid accumulation in primary mouse hepatocytes and livers in high-fat diet-fed mice.**
**a** Primary *mouse* hepatocytes were transfected with control siRNA (si-ctrl) or si-Maip1 at 50 nM. The mRNA levels were analyzed 48 h after transfection by qRT-PCRs, and the relative expression of *Maip1* was evaluated. $n = 5$ independent experiments. Results are presented as means ± SE. The p-value from Student's t-test and Cohen's d value are shown. **b** Primary *mouse* hepatocytes were transfected with the indicated mimics at 50 nM followed by co-staining with BODIPY493/503 for the lipid and Hoechst 33342 for the nuclei 7 days after transfection. Untransfected cells were treated with oleic acid (OA) at 50 μM for the last 48 h. Scale bars, 100 μm. **c** The relative amount of lipid was evaluated based on the images in (**b**). $n = 3$ biologically independent samples. Representative results from three independent experiments are presented as means ± SD. The p-value from Student's t-test and Cohen's d value are shown. **d** Triglycerides (TG) extracted from cells 7 days after transfection were quantitated. $n = 3$ independent experiments. Results are presented as means ± SE. The p-value from Student's t-test and Cohen's d value are shown. **e** and **f** Livers from mice fed a basic diet (BD) or a high-fat diet (HFD) for 8 weeks (**e**) or 16 weeks (**f**) were analyzed by qRT-PCR. $n = 6$ mice. Data are presented as means ± SE. p-values from Student's t-test and Cohen's d values are shown. Gray circles, BD; gray triangles, HFD. **g** Schematic presentation of the *miR-27b/MAIP1* axis-mediated hepatic lipid accumulation.

## Methods

**Cell culture**. Huh-7 and HepG2 cells (*human* hepatocarcinoma cell lines, obtained from the JCRB Cell Bank, Tokyo, Japan) and HEK293T cells (a transformed *human* embryonic kidney cell line) were cultured with Dulbecco's modified Eagle's medium (DMEM, 4.5 g/ml glucose; WAKO, Osaka, Japan) supplemented with 10% fetal bovine serum (FBS), streptomycin (100 μg/ml), and penicillin (100 U/ml). Cell cultures were maintained at 37 °C under 5% $CO_2$ and fresh culture medium was supplemented every 48 h.

**Plasmids**. Luciferase reporter plasmids were generated as follows. The genomic fragments containing the 3′UTRs of *B4GALT3* (nt 1800-2330), *MAIP1* (nt 1065-1235) and *PHPPL2* (nt 5074-5906) were amplified from genomic DNA of Huh-7 cells using the primers listed in Supplementary Table 4 and integrated between *XhoI* and *NotI* of psiCHECK2 (Promega, Madison, WI). Nucleotide substitutions (ACTGTGAA to ATTATAAA) were introduced into each seed sequence by the inverse PCR method[91], generating the mutant reporter plasmids. psiCHECK2-27bT was generated by cloning the double-stranded oligonucleotides complementary to the sequence of *miR-27b* into the *XhoI* and *NotI* sites. shRNA-expressing LV vectors, TRC1.0 Lentiviral Plasmid Vector pLKO.1-puro-shRNA (MISSION shRNA, Sigma Aldrich, St. Louis, MO), was obtained from the Center for Medical Research and Education, Osaka University. A negative control shRNA-expressing LV vector, TRC1.0 Lentiviral Plasmid Vector pLKO.1-puro-shRNA-control, was purchased from Sigma Aldrich. To generate B4GALT3- and MAIP1-expressing LV vectors, coding regions were amplified from the cDNA of Huh-7 cells using the primers listed in Supplementary Table 4 and cloned between *NotI* and *XbaI* of CSII-EF-MCS-IRES2-Venus[92]. Their *eukaryotic translation elongation factor (EF)-1α* promoter regions were replaced with the *thymidine kinase (TK)* promoter by cloning DNA fragments of the *TK* promoter region amplified from psiCHECK2 between *AgeI* and *XhoI*, generating CSII-TK-B4GALT3 and CSII-TK-MAIP1, respectively. CSII-TK-MCS was similarly generated using CSII-EF-MCS[93] as a control vector. CSII-EF-IRES2-Venus, CSII-EF-MCS, pCMV-VSVG-RSV-Rev and pCAG-HIVgp were provided by RIKEN BRC.

**miRNA and siRNA transfection experiments**. MirVana miRNA mimic negative control #1 (a random sequence mimic molecule), mirVana miRNA mimic hsa-miR-27b-3p (MC10750), silencer select siRNA negative control #1 (a siRNA with no sequence similarity to *mouse*, *rat*, or *human* genes), and silencer select pre-designed siRNA-*human B4GALT3*, -*human MAIP1* and -*human PHLPP2*, and -*mouse Maip1* were purchased from Thermo Fisher Scientific (Grand Island, NY). MiRNA and siRNA transfection was performed using Lipofectamine RNAiMAX reagent (Thermo Fisher Scientific) according to the manufacturer's instructions. Four hours after transfection, the culture medium was replaced with DMEM (1.0 g/ml glucose; Sigma Aldrich) supplemented with 10% FBS, streptomycin (100 µg/ml), and penicillin (100 U/ml).

**Quantitative reverse transcription-polymerase chain reaction (qRT-PCR)**. Huh-7 or HepG2 cells were seeded in 48-well plates at a density of $1 \times 10^4$ cells/well and transfected with miRNA mimics or siRNA at 10 nM the next day. For the overexpression experiments, cells were transduced with LV vectors right after seeding and transfected with miRNA mimics at 10 nM 24 h post-transduction. Total RNA was isolated using ISOGEN (Nippon Gene, Tokyo, Japan). Complementary DNA (cDNA) was synthesized using a Superscript VILO cDNA synthesis kit (Thermo Fisher Scientific) according to the manufacturer's instructions. Quantitative PCR (qPCR) was performed using a StepOnePlus real-time PCR system (Thermo Fisher Scientific) using THUNDERBIRD SYBR qPCR Mix (Toyobo, Osaka, Japan). The primers used in this study are listed in Supplementary Table 4. Values were normalized by GAPDH expression.

**Western blotting**. Protein samples were prepared 72 h after transfection using RIPA lysis buffer (Thermo Fisher Scientific) containing a proteinase inhibitor cocktail (Nacalai Tesque, Kyoto, Japan) and a phosphatase inhibitor cocktail (Nacalai Tesque). Whole-cell lysates (15–20 µg) were subjected to Western blotting. The following antibodies were used: rabbit anti-G3PDH polyclonal antibody (1:2000, Trevigen, Gaithersburg, MD); rabbit anti-MAIP1 polyclonal antibody (1:1000) (Atlas Antibodies, Stockholm, Sweden); rabbit anti-PHLPP2 polyclonal antibody (1:1000) (Abcam, Cambridge, UK); rabbit anti-B4GALT3 polyclonal antibody (1:500) (Atlas Antibodies); rabbit anti-AMPK polyclonal antibody (1:1000) (Cell Signaling Technology, Danvers, MA); rabbit anti-phosphorylated AMPK polyclonal antibody (1:1000) (Cell Signaling Technology). Signals were developed using Chemi-Lumi One Super (Nacalai Tesque) and visualized using ImageQuant LAS3000 (Fujifilm, Tokyo, Japan). Signal intensities were quantified using MultiGauge ver.3 software (Fujifilm). Samples prepared in three independent experiments were analyzed on the same membrane.

**Microarray analysis**. Huh-7 cells were transfected with control miRNA or *miR-27b* mimics at 10 nM, and total RNA samples were extracted 48 h post-transfection. Hybridization, signal acquisition, and data analysis were performed by Milteny Biotechnology (Bergisch Gladbach, Germany). Briefly, 100 ng of the total RNA sample was used to produce Cy3-labeled cRNA using the Agilent Low Input Quick Amp Labeling Kit (Agilent Technologies, Santa Clara, CA). Hybridization was performed using Agilent Whole Human Genome Oligo Microarrays 8x60K V2. Fluorescence signals were detected using Agilent's Microarray Scanner System (Agilent Technologies). All the microarray data in this work have been deposited at GEO under accession number GSE174075.

**Lipid droplet staining**. Huh-7 or HepG2 cells were seeded in 48-well plates at a density of $1 \times 10^4$ cells/well or $0.2 \times 10^4$ cells/well, respectively, and transfected with miRNA mimics or siRNA at 10 nM the next day. For overexpression

experiments, cells were transduced with LV vectors right after seeding and transfected with miRNA mimics at 10 nM 24 h post-transduction. As a control for each experiment, cells were cultured in the presence of 50 µM bovine serum albumin (BSA)-conjugated oleic acid (OA) for the last 48 h. The BSA-conjugated OA stock solution was prepared by vigorous agitation of 10 mM OA (Sigma Aldrich) in 20% fatty acid-free BSA (Wako)/PBS. Cells were fixed with 4% paraformaldehyde for 15 min at room temperature 96 h post-transfection. Lipid droplets were stained with 1 µg/mL of BODIPY 493/503 (Thermo Fisher Scientific) and cell nuclei were counterstained with 5 µg/mL Hoechst 33342 (Thermo Fisher Scientific) for 30 min at room temperature. Four images per well were taken by fluorescence microscopy (BZ-9000, Keyence, Osaka, Japan). Signal intensity was measured using a BZ-II analyzer (Keyence).

**Quantitation of cellular triglycerides (TGs)**. Huh-7 cells ($1.5 \times 10^5$ cells/well) and primary *mouse* hepatocytes ($1 \times 10^5$ cells/well) were seeded in 6- and 24-well plates and prepared for lipid extraction at days 4 and 7 of transfection, respectively. Cells were homogenized in the lysis buffer (1% Triton X-100/150 mM NaCl/1 mM EDTA/50 mM Tris–HCl, pH 7.5) and lipids were extracted according to the Folch method[94]. Briefly, 4 volume of chloroform (Fujifilm)/methanol (Nacalai Tesque) (1:2) was added to the homogenates and mixed well. After adding 1.25 volumes each of chloroform and distilled water and mixing, the lower phase was separated by centrifugation. The collected lower phase was dried-up and the residual lipids were resolved in 2-propanol (Nacalai tesque). The TG levels were measured using the enzymatic assay kit Cholestest TG (Sekisui Medical, Tokyo, Japan). TG values were normalized by the protein concentration of the homogenates.

**Primary *mouse* hepatocytes**. Hepatocytes were isolated from perfused livers of 4–9 week-old C57B/6N mice as previously described[95]. Briefly, livers in anesthetized mice were serially perfused with the warmed pre-perfusion buffer (Hank's balanced salt solution (HBSS, Thermo Fisher Scientific)/0.5 mM EDTA (Dojindo, Kumamoto, Japan)/25 mM HEPES (Nacalai tesque)) and the perfusion buffer (0.5 µg/ml Collagenase D (Sigma Aldrich)/HBSS with Ca$^{2+}$, Mg$^{2+}$ (Thermo Fischer Scientific)/25 mM HEPES) via the inferior vena cava. Hepatocytes were purified by repeated centrifugation at $50 \times g$ for 5 min and viable cells were further purified by centrifugation in a Percol solution (45% Percol plus (GE Healthcare Bio-Sciences AB, Uppsala, Sweden)/5% 10xHBSS/50% DMEM) at $100 \times g$ for 20 min. Purified hepatocytes were plated in a collagen-coated 96-well and a 24-well plate (Iwaki Glass, Tokyo, Japan) at a density of $2.5 \times 10^4$ and $1 \times 10^5$ cells/well, respectively. After three-hour culture in William's medium E (Thermo Fisher Scientific) supplemented with 10% FBS, streptomycin (100 µg/ml), penicillin (100 U/ml) and insulin (5 µg/ml, WAKO), the culture medium was replaced with Cellartice Power Primary HEP Medium (Takara Bio, Otsu, Japan). Cells were transfected with miRNA mimics or siRNAs at 50 nM using RNAiMAX on the next day of seeding. RNA preparation was performed on day 2 and staining with BODIPY493/503 and TG preparation was performed on day 7 of transfection.

**Mice fed a high fat-diet (HFD)**. Six-week-old male C57B/6J mice were fed a standard chow (Oriental Yeast Co., Osaka, Japan) or an HFD (HFD32; 5.1 kcal/g, 23% kcal from carbohydrate, 57% kcal from fat, 20% kcal from protein; CLEA Japan Inc., Tokyo, Japan) for either 8 or 16 weeks.

**Luciferase assay**. Huh-7 cells ($1 \times 10^4$/well) were seeded in 96-well plates on the day before transfection. The cells were co-transfected with 25 ng of reporter plasmids carrying wild-type or mutant 3′ UTRs and miRNA mimics (10 nM) using Lipofectamin3000 (Thermo Fisher Scientific) according to the manufacturer's instructions. The luciferase activity was measured 48 h post-transfection using the dual-luciferase reporter assay (Promega) according to the manufacturer's instructions. Luminescence was measured using a luminometer (Orion II Microplate Luminometer, Berthold Detection Systems, Pforzheim, Germany). All assays were performed in triplicate, and three independent experiments were evaluated.

**Lentiviral vectors**. LV vectors were prepared according to the RIKEN BRC protocol (http://cfm.brc.riken.jp/). Briefly, LV vector plasmids, together with pCMV-VSVG-RSV-Rev and pCAG-HIVgp, were transfected into HEK293T cells using polyethylenimine transfection reagents (Polysciences, Warrington, PA), and viral particles were purified from the culture supernatants by ultracentrifugation ($50,000 \times g$, 2 h). Vector titers were determined by transducing Huh-7 cells and subsequent genomic qPCR using CSII-EF-MCS-IRES2-Venus as a standard.

**Animal experiments**. All procedures involving animals and their care were approved by the Osaka University Institutional Animal Care and Use Committee in accordance with Fundamental Guidelines for Proper Conduct of Animal Experiment and Related Activities in Academic Research Institutions under the jurisdiction of the Ministry of Education, Culture, Sports, Science and Technology of Japan. We have complied with all the ethical regulations for animal experiments related to the approval.

**Statistics and reproducibility**. Statistical analyses were performed using Graph-Pad Prism 5.04 or 9.5.1. For quantitation of mRNA expression in Figs. 4a, 5a, 5e and Supplementary Fig. 1c, and lipid accumulation in Fig. 1e and Supplementary Fig. 1b, data from biologically independent samples were plotted as means ± SD. For the mouse and biopsy analysis, data from individuals were plotted as means ± SE. For the rest analyses, data from independent experiments were plotted as means ± SE. The exact sample size for each graph was presented in the corresponding figure legend. Comparisons between two groups and multiple groups were performed using unpaired two-tailed Student's $t$-test and one-way ANOVA followed by Bonferroni's post-hoc comparisons tests, respectively. $P < 0.05$ was considered to indicate significance. For effect size estimation, Cohen's $d$ and Pearson's $r$ values were calculated for the Student's $t$-tests and the multiple-comparison tests, respectively.

**Reporting summary**. Further information on research design is available in the Nature Portfolio Reporting Summary linked to this article.

## Data availability

The Data that supports the findings of this study are included in this article and its supplementary files. The source data behind the graph in the paper is available in Supplementary Data 3. Uncropped scans are available in Supplementary Fig. 5. Data from microarray analysis have been deposited in GEO under accession number GSE174075. Included data IDs are GSM5285801, GSM5285802, GSM5285803 GSM5285804 and GSM5285805. The accesion code for GSE174075 is obkxakwovbcxvcv. The following plasmids are available from Addgene: CSII-TK-MCS (#203597); CSII-TK-B4GALT3 (#203598); CSII-TK-MAIP1 (#203599); psicheck2-27bT (#203600); psicheck2-B4GALT3 (#203601); psicheck2-B4GALT3mut (#203602); psicheck2-MAIP1 (#203603); psicheck2-MAIP1mut (#203604); psicheck2-PHLPP2 (#203605); psicheck2-PHLPP2mut (#203606).

## Code availability

The Agilent Feature Extraction Software (Agilent Technologies) was used to read out and process the microarray image files. The data files were further analyzed using Rosetta Resolver gene expression data analysis system (Rosetta Biosoftware, Seattle, USA). DAVID 2023 was used for the pathway analysis in Supplementary Fig. 2 and is available at https://david.ncifcrf.gov.

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

## Acknowledgements
This research was financially supported by the Japan Society for the Promotion of Science (JSPS) KAKENHI (grant numbers 18K06656, 21K18247) and by Bristol-Myers Squibb Foundation.

## Author contributions
E.S. designed experiments, collected and analyzed data and wrote the manuscript. T.I., R.S., M.T.-G. and T.F. collected and analyzed data. F.S. supervised the study. H.M. supervised the study and edited the manuscript.

## Ethics declarations
Animal experiments.

## Competing interests
The authors declare no competing interests.
