## [Peer Review File · Communications Biology]

Reviewers' comments:

Reviewer #1 (Remarks to the Author):

Dear authors,

In the manuscript, the authors showed the novel target of miR-27b on the lipid accumulation in hepatic cells. The findings and contents presented in the manuscript are interesting. However, there are several issues that should be resolved.

Major comments:

1) The authors clearly showed that MAIP1 is a novel target of miR-27b on the lipid accumulation in this manuscript. However, it is still unclear why the decrease in MAIP-1 causes the lipid accumulation in hepatic cells. The exact mechanisms of decreased MAIP1-mediated lipid accumulation have not shown in the manuscript, although the authors showed that AMPK is not involved in. Therefore, it is better to show the potential mechanisms of MAIP1-mediated suppression of lipid accumulation in hepatic cells, although the data are preliminary.

2) The authors clearly showed that MAIP1 is a novel target of miR-27b on the lipid accumulation in "cultured" hepatic cells. Such data are solid and interesting, but only cultured cells. Confirmation about the direct effect of decreased MAIP1 on the lipid accumulation in vivo situations are not investigated in this manuscript. In fact authors showed the decrease in MAIP1 on high-fat diet mice. However, investigations of direct loss or decrease in MAIP1 in vivo situation were not performed in this manuscript. Conditional KO or in vivo-siRNA technic are important to show the direct effect of MAIP1 on lipid accumulation in vivo situation.

3) Related to the above comments, I think the clinical approach might be a good supportive data to prove the importance of MAIP1 on the lipid accumulation in liver, if the in vivo animal experiment are difficult. Liver biopsy samples of NAFLD patients are useful to investigate the decrease in MAIP1 during the steatosis. Are there any supportive data about the human steatosis samples?

4) PPAR γ is well known the direct target of miR-27b, and is involved in lipid accumulation. The authors excluded the PPAR γ -mediated pathways in this manuscript. For me, explanations and steps are still unclear for the description of the manuscript. Is it adequate to exclude the effect of PPAR γ from all of steps to investigate the lipid accumulation? I think, at least, the data about the level (or function) of PPAR γ should be shown when miR-27b is knockdown.

5) The authors used fluorescent dyes, BODIPY493/503 for the evaluation of lipid accumulation in this manuscript. However, it is not adequate to use only imaging analysis of fluorescent signal as quantitative evaluation of lipid accumulation. I can not see the difference of several fluorescent photos (e.g. Figure 3C, Figure 5B). Other quantitative analysis (such as TG contents) should be performed to confirm the effects.

6) The authors concluded that MAIP1, but not B4GALT3, ameliorated miR-27b-induced lipid accumulation. However, is it adequate to assert so? The tendency of the decrease in lipid accumulation was observed when LV-B4G3 treated in Figure 5F. If the sample numbers are increased, statistically significance will be observed?

Minor comments:

1) Schematic illustrations are useful to show the miR27b-MAIP1-mediated pathways on lipid accumulation in hepatic cells.

Reviewer #2 (Remarks to the Author):

In this manuscript, the authors have succeeded to identify a novel miR-27b-dependent lipid regulatory pathway through targeting MAIP1. The experimental designs were well polished. Manuscript has been also well-written. This reviewer has several comments as described below:

Major comments

#1. The authors have referred too much review articles. Original articles should be referred in this manuscript as far as possible.

#2. In introduction, L52-55, the authors mentioned "NAFLD is considered a risk factor of more progressive diseases". It is better to refer appropriate references. If there are no references, the authors should correct it into a more suitable expression.

#3. The authors employed lipid droplet assay, but the correlation between image data and value are differed substantially from each experiment. Are there any controls occurring lipid droplets induced by adipocyte differentiation reagents? These controls help to understand the degree of contribution of target genes to lipid droplets.

#4. In GO enrichment analysis, barplot or dotplot might be better than Table S3.

Minor comments

#1. In introduction, L79-80, references are needed.

#2. In annotation of figure 2C and D, sh-ctrl/target are better than ctrl/KD. Are there any difference "relative area" and "relative lipid accumulation"?

#3. In figure 4D, Gene order should be unified to other figures.

#4. Could you check statistical analysis of figure 5A and D?

#5. In annotation of Figure6D and E, HFD-8w/16w are better than only HFD.

#6. In addition to PPARG, could you add the discussion about effect of miR-27b on major lipid metabolism-related pathways based on microarray data.

#7. There are some machine-dependent characters and typos.

According to your helpful suggestions, we modified the manuscript and figures as bellow. The sentences we revised to manuscript are shown in red. We also replaced some review articles to original ones according to reviewer #2's comment and the replaced or newly inserted references are highlighted with a blue color.

To reviewer #1,

Thank you for your review and valuable comments.

Major comments:

1) The authors clearly showed that MAIP1 is a novel target of miR-27b on the lipid accumulation in this manuscript. However, it is still unclear why the decrease in MAIP-1 causes the lipid accumulation in hepatic cells. The exact mechanisms of decreased MAIP1-mediated lipid accumulation have not shown in the manuscript, although the authors showed that AMPK is not involved in. Therefore, it is better to show the potential mechanisms of MAIP1-mediated suppression of lipid accumulation in hepatic cells, although the data are preliminary.

Response: We are currently investigating the mechanisms underlying the decreased MAIP1-mediated lipid accumulation. Although we had described our hypothesis in the original Discussion section, it would require a substantially large and complex analysis to prove. We must therefore defer a full consideration of this question until a future study.

2) The authors clearly showed that MAIP1 is a novel target of miR-27b on the lipid accumulation in "cultured" hepatic cells. Such data are solid and interesting, but only cultured cells. Confirmation about the direct effect of decreased MAIP1 on the lipid accumulation in vivo situations are not investigated in this manuscript. In fact authors showed the decrease in MAIP1 on high-fat diet mice. However, investigations of direct loss or decrease in MAIP1 in vivo situation were not performed in this manuscript. Conditional KO or in vivo-siRNA technic are

important to show the direct effect of MAIP1 on lipid accumulation *in vivo* situation.

Response: As you pointed out, an analysis of conditional MAIP1 KO would be extremely helpful to show the role of MAIP1 in *in vivo* lipid accumulation. We would like to work on this in the very near future as an independent report.

3) Related to the above comments, I think the clinical approach might be a good supportive data to prove the importance of MAIP1 on the lipid accumulation in liver, if the *in vivo* animal experiments are difficult. Liver biopsy samples of NAFLD patients are useful to investigate the decrease in MAIP1 during the steatosis. Are there any supportive data about the human steatosis samples?

Response: We agree with you, but unfortunately we are not in a position to handle liver biopsies by ourselves. Instead, we analyzed a pre-existing RNA-seq database of liver samples from 93 NASH patients (GSE225740). This new analysis revealed a downward trend in MAIP1 expression during the progression of steatohepatitis (Figure S4), which we discuss in the revised manuscript (lines 299-305).

4) PPARG is well known the direct target of miR-27b, and is involved in lipid accumulation. The authors excluded the PPARG-mediated pathways in this manuscript. For me, explanations and steps are still unclear for the description of the manuscript. Is it adequate to exclude the effect of PPARG from all of steps to investigate the lipid accumulation? I think, at least, the data about the level (or function) of PPARG should be shown when miR-27b is knockdown.

Response: We appreciate your concerns, but a number of studies in the literature indicate that, in contrast to in the adipose tissue, PPARG promotes lipid accumulation in the liver (Yu S *et al.*, J Biol Chem. 278 (2003) 498–505; Yamazaki T *et al.*, J Nutr Biochem. 22 (2011) 543–53; Matsusue K *et al.*, J. Clin. Invest. 111 (2003) 737–747; Wolf Greenstein A *et al.*, J. Endocrinol. 232 (2017) 107–121). From this perspective, we believe that excluding PPARG from our experiments, such as for identifying novel genes involved in suppression of lipid accumulation in hepatic cells, is appropriate. We added a description in the manuscript to help readers understand our rationale (lines 166-172).

5) The authors used fluorescent dyes, BODIPY493/503 for the evaluation of lipid accumulation in this manuscript. However, it is not adequate to use only imaging analysis of fluorescent signal as quantitative evaluation of lipid accumulation. I can not see the difference of several fluorescent photos (e.g. Figure 3C, Figure 5B). Other quantitative analysis (such as TG contents) should be performed to confirm the effects.

Response: Thank you for pointing this out. To address this issue, we extracted lipids from cells, evaluated TG contents and presented the results along with the fluorescence quantitation (Figures 1C, 1F, 3E, 5D and 5H; lines 120-122, 125-126, 202-205 and 254-255)

6) The authors concluded that MAIP1, but not B4GALT3, ameliorated miR-27b-induced lipid accumulation. However, is it adequate to assert so? The tendency of the decrease in lipid accumulation was observed when LV-B4G3 treated in Figure 5F. If the sample numbers are increased, statistically significance will be observed?

Response: According to your comment, we increased the number of experiments in the LV-B4G3 and LV-MAIP1 analyses to six each, and obtained the same conclusion as in the previous analysis (Figure 5G and 5H; lines 1088-1089).

Minor comments:

1) Schematic illustrations are useful to show the miR27b-MAIP1-mediated pathways on lipid accumulation in hepatic cells.

Response: We added a brief schematic illustration (Figure 6F).

To reviewer #2,

We thank you for your review and valuable comments.

Major comments

#1. The authors have referred too much review articles. Original articles should be referred in this manuscript as far as possible.

Response: According to your comments, we replaced some review articles to

original ones. The replaced or newly inserted references were highlighted with blue color.

#2. In introduction, L52-55, the authors mentioned “NAFLD is considered a risk factor of more progressive diseases”. It is better to refer appropriate references. If there are no references, the authors should correct it into a more suitable expression.

Response: We modified the description in the manuscript in lines 53-56.

#3. The authors employed lipid droplet assay, but the correlation between image data and value are differed substantially from each experiment. Are there any controls occurring lipid droplets induced by adipocyte differentiation reagents? These controls help to understand the degree of contribution of target genes to lipid droplets.

Response: According to your helpful comments, we analyzed cells treated with oleic acids in parallel, to provide a control for comparison in each experiment (Figures 1A, 1B, 1D and 1E; 3C and 3D; 5F, 5G, 5B and 5C; 6B and 6C; lines 970-971, 977-978, 1024-1025, 1085-1086 and 1104).

#4. In GO enrichment analysis, barplot or dotplot might be better than Table S3.

Response: According to your comment, we prepared barplots for pathway analysis of up- and down-regulated genes (Figure S2). We also added a list of lipid metabolism-related up-regulated genes to the new Table S3 and a corresponding explanatory passage in the text (lines 137-157).

Minor comments

#1. In introduction, L79-80, references are needed.

Response: We added reference (line 71).

#2. In annotation of figure 2C and D, sh-ctrl/target are better than ctrl/KD. Are there any difference “relative area” and “relative lipid accumulation”?

Response: The annotations were changed to sh-ctrl/target and the phrase “relative area” was standardised to “relative lipid accumulation” throughout

(Figures 2C and D).

#3. In figure 4D, Gene order should be unified to other figures.

Response: According your comment, we changed the gene order (Figure 4D).

#4. Could you check statistical analysis of figure 5A and D?

Response: Statistical analysis was performed and presented in figures 5A and E as well as their figure legends.

#5. In annotation of Figure6D and E, HFD-8w/16w are better that only HFD.

Response: The annotations were changed following your comment (Figure 6E).

#6. In addition to PPARG, could you add the discussion about effect of miR-27b on major lipid metabolism-related pathways based on microarray data.

Response: We added some descriptions in the manuscript (lines 137-157).

#7. There are some machine-dependent characters and typos.

Response: We apologize for this and corrected them.

REVIEWERS' COMMENTS:

Reviewer #1 (Remarks to the Author):

To the authors,

The revised manuscript is dramatically improved. Our comments have been carefully answered. We have no more comments.

Sincerely yours,

Reviewer #2 (Remarks to the Author):

The authors addressed the major issues and significantly improved the revised version of the manuscript.